# Hierarchical regulation of autophagy during adipocyte differentiation

**Mahmoud Ahmed**[ORCID], **Trang Huyen Lai**[ORCID], **Trang Minh Pham, Sahib Zada**[ORCID], **Omar Elashkar**[ORCID], **Jin Seok Hwang**[ORCID], **Deok Ryong Kim**[ORCID] *

Department of Biochemistry and Convergence Medical Sciences and Institute of Health Sciences, Gyeongsang National University, College of Medicine, Jinju, South Korea

* drkim@gnu.ac.kr

## Abstract

We previously showed that some adipogenic transcription factors such as CEBPB and PPARG directly and indirectly regulate autophagy gene expression in adipogenesis. The order and effect of these events are undetermined. In this study, we modeled the gene expression, DNA-binding of transcriptional regulators, and histone modifications during adipocyte differentiation and evaluated the effect of the regulators on gene expression in terms of direction and magnitude. Then, we identified the overlap of the transcription factors and co-factors binding sites and targets. Finally, we built a chromatin state model based on the histone marks and studied their relation to the factors' binding. Adipogenic factors differentially regulated autophagy genes as part of the differentiation program. Co-regulators associated with specific transcription factors and preceded them to the regulatory regions. Transcription factors differed in the binding time and location, and their effect on expression was either localized or long-lasting. Adipogenic factors disproportionately targeted genes coding for autophagy-specific transcription factors. In sum, a hierarchical arrangement between adipogenic transcription factors and co-factors drives the regulation of autophagy during adipocyte differentiation.

## Introduction

Previous studies suggested one-to-one interactions between adipogenic transcription factors and autophagy. CEBPB transactivates *Atg4b*, a key protein in the autophagy machinery [1]. The activation of autophagy through this pathway relieves the repression of adipogenic activators such as PPARG. FOXO1, a transcription factor with several autophagy targets, was suggested to the repress *Pparg* gene in the presence of insulin sensitizers [2]. This repression is likely to be lifted in early adipogenesis.

A previous study from our laboratory showed that autophagy gene products are regulated as part of the transcription program of adipogenesis [3]. This regulation is achieved through adipogenic transcription factors PPARG and CEBPB either directly or indirectly through autophagy specific factors. The magnitude and the ordering of this regulation remain to be investigated. The key questions in this regard are when and where the binding of those two

**Data Availability Statement:** The datasets analyzed in this study can be found in online repositories. The names of the repository and accession numbers can be found below: Figshare: https://doi.org/10.6084/m9.figshare.9906182,

https://doi.org/10.6084/m9.figshare.9906200, and https://doi.org/10.6084/m9.figshare.13088624.

**Funding:** This study was supported by the National Research Foundation of Korea (NRF) grant funded by the Ministry of Science and ICT (MSIT) of the Korean government [2015R1A5A2008833 and 2020R1A2C2011416].

**Competing interests:** The authors have declared that no competing interests exist.

factors occurs on the target genes in respect to each other and to their co-factors. In addition, does these regulatory links favor particular kinds of autophagy sub-functions and/or down stream effectors.

Here, we used gene expression and DNA-binding data to model the transcription factor and co-factors binding events during differentiation and their effect on autophagy genes. We used histone modification data to correlate these events with chromatin states. A hierarchical arrangement of known adipogenic transcription factors and co-factors emerged in the regulation of autophagy during adipogenesis. We evaluated the spatial and temporal aspects of this arrangement. These included the factors' contributions to gene expression, the dependency between regulators, the reliance on chromatin states, and the type of binding targets.

## Materials and methods

### Expression & binding data in differentiating adipocytes

We collected two datasets of RNA-seq and ChIP-seq of 3T3-L1 pre-adipocytes, which were induced to differentiate using 3-isobutyl-1-methylxanthine, dexamethasone, and insulin (MDI) and sampled at different time points [4]. We curated the samples' metadata using a unified language across the studies and processed the raw data using standard pipelines. The processed gene expression data were made available as a Bioconductor data package (curatedAdipoRNA). The data are presented as gene counts at different time points (0 to 240 hr) (Table 1). The processed DNA-binding data of transcription factors, co-factors, and histone modifications were made available as a similar package (curatedAdipoChIP). Data are presented in this package as the reads count in a consensus peak set (Tables 2 & 3). Moreover, we provided links to the identified peaks as well as the signal tracks files. The packages also document the pre-processing and processing pipelines.

### Expression data of genetically perturbed adipocytes

We obtained two gene expression datasets of *Cebpb* (RNA-seq) and *Pparg*-knockdown (microarrays) from matching MDI-induced 3T3-L1 pre-adipocytes time-course experiments

**Table 1. MDI-induced 3T3-L1 gene expression data by RNA-seq.**

| GEO ID | N | Time (hr) | Ref. |
|---|---|---|---|
| GSE100056 | 2 | 24 | [32] |
| GSE104508 | 3 | 192 | [33] |
| GSE35724 | 3 | 192 | [34] |
| GSE50612 | 4 | 0/144 | [35] |
| GSE50934 | 6 | 0/168 | [36] |
| GSE53244 | 3 | 0/48/240 | [37] |
| GSE57415 | 4 | 0/4 | [38] |
| GSE60745 | 12 | 0/24/48 | [39] |
| GSE64757 | 6 | 168 | [40] |
| GSE75639 | 3 | 0/48/168 | [41] |
| GSE84410 | 5 | 0/4/48 | [42] |
| GSE87113 | 5 | 0/2/4/48/168 | [43] |
| GSE89621 | 3 | 240 | [44] |
| GSE95029 | 8 | 0/48/144/192 | [45] |
| GSE95533 | 10 | 4/0/24/48/168 | [46] |
| GSE96764 | 6 | 0/2/4 | [47] |

**Table 2. Transcription factors binding data.**

| SRA ID | N | Antibody | Ref. |
|---|---|---|---|
| SRP000630 | 12 | PPARG/ RXRG | [48] |
| SRP002337 | 2 | PPARG | [49] |
| SRP002507 | 2 | CEBPB | [50] |
| SRP006001 | 9 | CEBPB/ CEBPD/ RXRG/ PPARG | [51] |
| SRP028367 | 3 | PPARG/ MED1 | [52] |
| SRP041249 | 3 | RXRG/ MED1/ EP300 | [53] |
| SRP100871 | 28 | CTCF/ MED1/ NCOR1/ EP300 | [46] |

**Table 3. Histone modification data.**

| SRA ID | N | Antibody | Ref. |
|---|---|---|---|
| SRP002337 | 11 | H3K4me3/ H3K27me3/ H3K36me3/ H3K4me2/ H3K4me1/ H3K27ac | [49] |
| SRP041249 | 6 | H3K27ac/ H3K4me1/ H3K4me2 | [53] |
| SRP064188 | 3 | H3K27me3/ H3K9me3 | [54] |
| SRP078506 | 6 | H3K4me3 | [42] |
| SRP100871 | 6 | H3K27ac/ H3K4me1/ H3K4me2 | [46] |

(Table 4). The knockdown conditions were generated using shRNA or siRNA against *Cebpb* or *Pparg* respectively, or against scrambled sequences as controls. Gene counts and probe intensities were downloaded using GEOquery and used to quantify the gene expression from RNA-seq and microarray data, respectively [5].

## Mouse genome annotations

Mouse gene ontology (GO) terms of biological processes were used to identify the gene products relevant to autophagy and lipogenesis [6]. The Bioconductor package org.Mm.eg.db and GO.db were used to access the GO annotations [7, 8]. The gene accessor IDs were mapped between gene symbols and Entrez IDs using TxDb.Mmusculus.UCSC.mm10.knownGene [9]. The same package was used to extract gene coordinates in the mouse genome.

The terms "autophagy" (GO:0006914) and "lipid metabolic process" (GO:0006629) represented the genes involved in the process of autophagy and lipogenesis. "Negative" (GO:0010507) and "positive regulation" (GO:0010508) terms represented the genes involved in the regulation of autophagy. Autophagy was further broken down into selective forms (GO:0061912) gene sets such as "aggrephagy", "mitophagy" and "reticulophagy" or subtypes such as chaprone-mediated, late-endosomal microautophagy. GO terms for molecular functions were used to identify the functional categories of the transcription factors targets: "transcription" (GO:0003700), "kinases" (GO:0050222) and "phosphatases" (GO:0016791).

**Table 4. Perturbed MDI-induced 3T3-L1 gene expression data by RNA-seq.**

| GEO ID | N | KD | Ref. |
|---|---|---|---|
| GSE57415 | 8 | *Cebpb* | [38] |
| GSE12929 | 18 | *Pparg* | [55] |

## Differential gene expression

RNA-seq reads were aligned to the mm10 mouse genome and counted in known genes using HISAT2, and featureCount [10, 11]. Gene counts were filtered, normalized, transformed, and subjected to batch effects removal. Microarray probe intensities were filtered and collapsed to corresponding known genes, normalized, and transformed. To identify gene expression changes over time or in response to the knockdown of a gene, we applied differential gene expression analysis using DESeq2, or LIMMA [12, 13]. Briefly, the gene counts or the probe intensities were compared between conditions (#hr vs. 0 hr or knockdown vs. control). Fold-change and p-value for every gene in each comparison were calculated. False-discovery rate (FDR) was used to adjust for multiple testing.

## Binding peaks analysis

ChIP-seq reads were aligned to the mm10 mouse genome using BOWTIE2 [14]. Binding peaks were identified using MACS2 with the annotation file of the same genome [15]. Peaks were annotated and assigned to the nearest gene using ChIPSeeker [16]. The numbers of binding sites and targets were calculated in each sample. When more than one sample was available for a given ChIP antibody, only replicated binding sites or targets were included. The intersections of binding sites and targets among the samples were calculated and visualized using ggupset.

## Hidden Markov chain models

Multi-state hidden Markov chain models of transcription factors and histone modifications in differentiating adipocytes were built using ChromHMM [17]. Briefly, aligned ChIP-seq reads were binarized to 100/200 bp windows over the mm10 mouse genome. Multivariate hidden Markov chains were used to model the factor/marker's presence or absence in combinatorial and spatial patterns (states). Emission and transition probabilities for the states were calculated to express the probability of each factor/marker being in a given state and the probability of the states transitioning to/from another at different time points. State enrichment over genomic locations and around the transcription start sites was calculated. The R package segmentr (under development) was used to call ChromHMM, read, and visualize the output.

## Gene set enrichment and over-representation

To calculate GO terms' enrichment scores at different times of differentiation, we ranked all genes by fold-change, performed a walk of the gene set members over the ranked list, and compared it to random walks. The enrichment score is the maximum distance between the gene set and the random walk [18]. ChromHMM calculates the enrichment of states as $(C/A)/(B/D)$ where A is the number of bases in the state, B is the number of bases in external annotation, C is the number of basses in the state, and the annotation and D is the number of bases in the genome. clusterProfiler calculates the over-representation as the number of items in the query and subject groups compared to the groups' total number [19].

## Software & reproducibility

The analysis was conducted in R language and environment for statistical computing and graphics [20]. Several Bioconductor packages were used as data containers and analysis tools [21]. The software environment was packaged as Docker image (https://hub.docker.com/r/bcmslab/hierarchy). The code for reproducing the analysis and generating the figures and

tables in this manuscript is released under GPL-3 open source licence (https://github.com/BCMSLab/hierarchical_autophagy_regulation).

## Results

### Adipogenic factors regulate autophagy genes during differentiation

To examine the expression of autophagy genes during adipogenesis, we used a dataset of MDI-induced 3T3-L1 pre-adipocytes sampled at different time points and profiled by RNA-Seq. In addition, we used two datasets of a similar time-courses with *Cebpb* or *Pparg* perturbations. We found that pre-adipocytes responded to MDI induction by changes in gene expression as early as 4 hours and continued for days. The size of the response was reasonably stable during the differentiation and was evenly split (25% at a false-discovery rate (FDR) $< 0.2$) between genes regulated in either direction. The response was strong for adipogenesis and lipogenesis genes. A larger fraction (30% at day 2 and 50% at day 7 at FDR $< 0.2$) of the genes involved in these processes were progressively induced (up-regulated) up until day 7 of differentiation. We compared this pattern to that of lipogenic genes.

The autophagy response to the MDI induction is bi-phasic with an inflection point around day 2 (Fig 1A). The initial response involved the down-regulation of most autophagy genes ($> 40\%$ at FDR $< 0.2$). This pattern was reversed in the latter days, where many more autophagy genes were up-regulated (40% at day 7 at FDR $< 0.2$). At the gene set level, the products in the gene ontology (GO) term "autophagy" were represented in the down-regulated set (normalized enrichment score (NES) $< -1.3$) up to day 2 and in the up-regulated set (NES $> 0.8$) from then onward (Fig 1B). By contrast, the gene products in the GO term "lipid metabolic process" were always represented in the up-regulated (NES $> 1.2$) ranks in the list of genes. Although the trend is clear, the score at 24 hours appears to be an anomaly (p-value $> 0.05$), possibly because the differential expression at this time point is based on a small number of samples.

Adipogenic transcription factors such as CEBPB and PPARG drive autophagy gene expression changes. The expression of the genes coding for those two transcription factors was induced ($\log_2$ fold-change (FC) $> 1.75$ and 2.5 at day 2 at FDR $< 0.01$, respectively). The knockdown of these factors in pre-adipocytes produced a wide dysregulation of autophagy genes (Fig 1C). *Pparg*-knockdown resulted in the up-regulation of 5 to 30 (FDR $< 0.2$) autophagy genes during the first 48 hours of MDI-induction. More than ten autophagy genes were down-regulated (FDR $< 0.2$) by the factor knockdown in the later stages of differentiation. *Cebpb*-knockdown, on the other hand, resulted in the down-regulation (25 to 15 genes at FDR $< 0.2$) of autophagy genes in pre-adipocytes four hours after MDI induction. Overall, PPARG explains more of the variance in autophagy gene expression (3.5%) than CEBPB (2%) or co-factors (Fig 1D).

### Selective and organelle-specific autophagy exhibit stage-dependent activation

In agreement with the previous literature, we observed the induction of CEBPB and PPARG in early and intermediate adipogenesis. However, we could identify binding sites for both factors at all time points and in pre-adipocytes. The targets of CEBPB seem to be regulated for a brief period of time that coincided with the induction of the *Cebpb* expression and doesn't last for long (Fig 2B). By contrast, PPARG binding induced its targets' expression, and the induction lasted till the end of the experiment (Fig 2A).

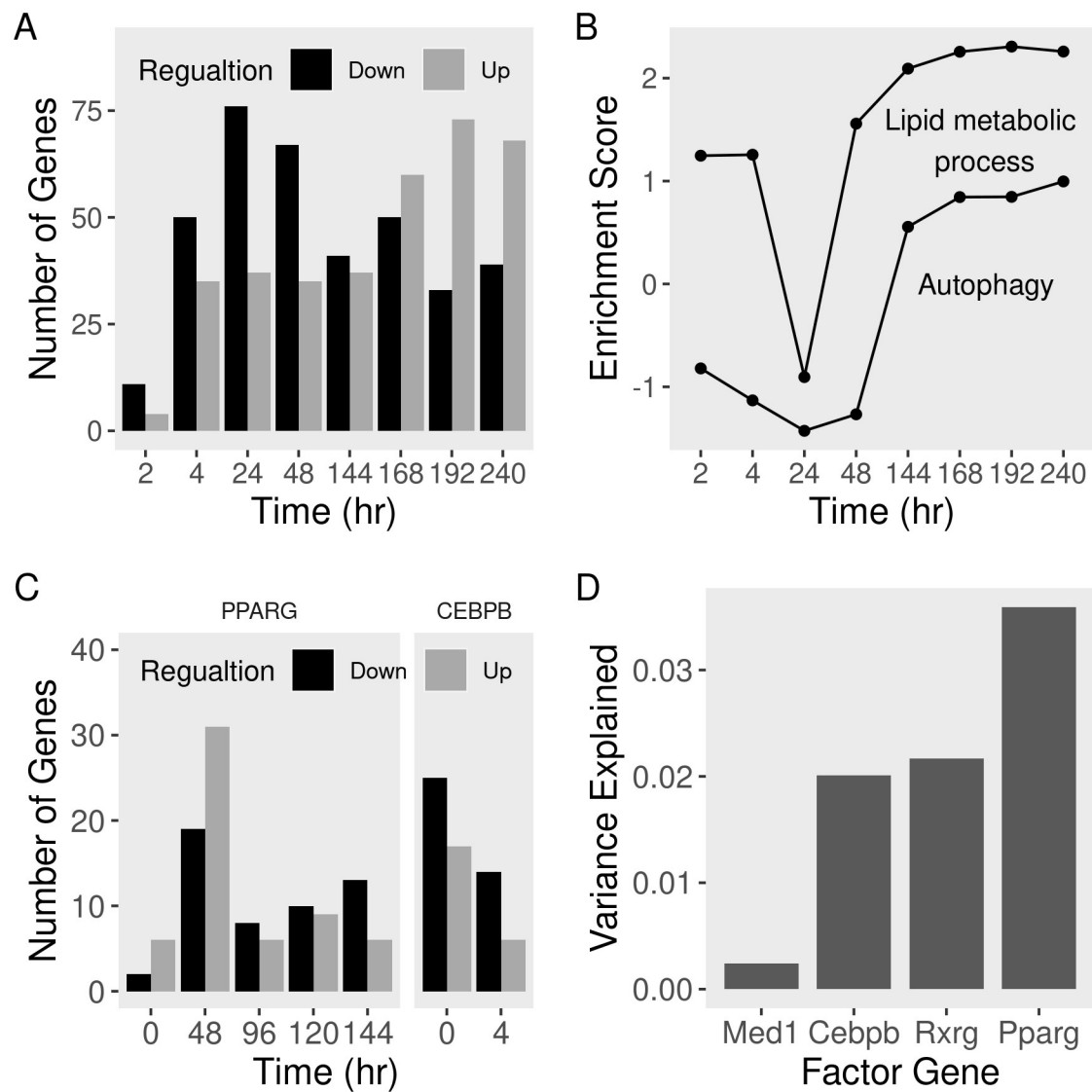

**Fig 1. Expression of autophagy gene products during adipocyte differentiation.** We curated a dataset of MDI-induced 3T3-L1 pre-adipocytes gene expression using RNA-seq publicly available data (Table 1). Read counts were used to quantify the expression of autophagy (and lipogenesis) genes at different times points of differentiation. Gene expression was compared to pre-adipocytes (0 hr) to calculate fold-change and p-values. Genes were descendingly ranked by fold-change. A) Number of up-or down-regulated genes in the non-modified course. B) Over-representation of the "autophagy" and "lipid metabolic process" term members in the top or bottom ranks of the list. We obtained two datasets of genetically perturbed (shRNA/siRNA against scrambled sequences (control) or *Cepbp and Pparg) 3T3-L1 differentiation courses* (Table 4). Gene expression was profiled using RNA-seq and microarrays, respectively. Differential expression was used to calculate the difference in the read counts and probe intensities between knockdown and controls at the corresponding time points. C) Number of up-or down-regulated genes with *Pparg*-or *Cepbp*-perturbations. D) Fraction of the variance of autophagy genes explained by the expression of adipogenic regulators coding genes.

To further explore the effect of the factor binding on autophagy, we calculated the enrichment scores of several autophagy-related terms at different time points of differentiation (Fig 2C). The term "negative regulation of autophagy" was enriched in the down-regulated genes in the first two days of differentiation. This was reversed after 48 hours. Besides, the positive regulation term was later enriched in up-regulated genes. Organelle-specific autophagy terms

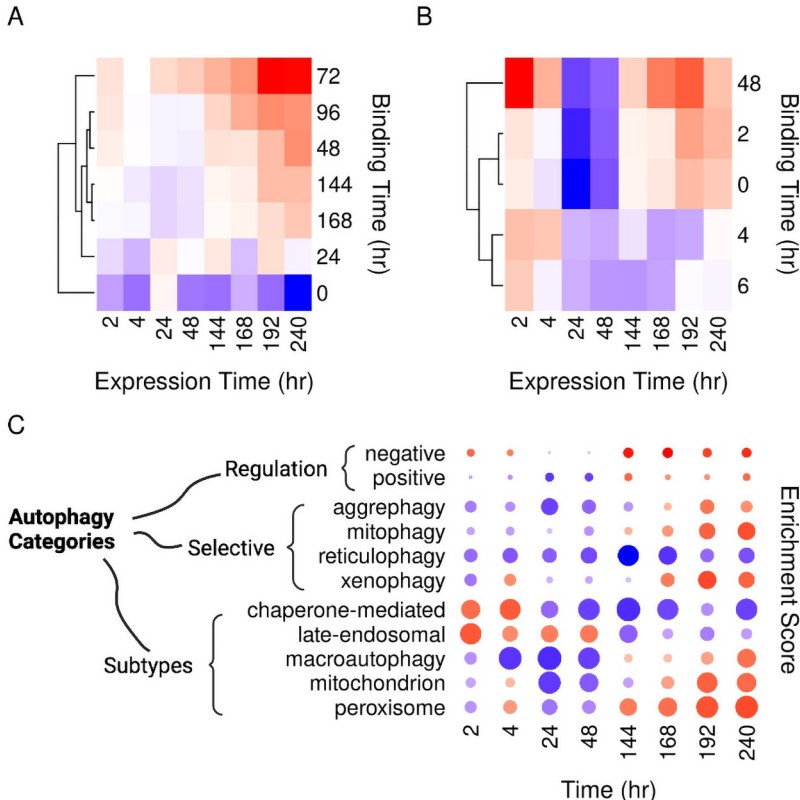

**Fig 2. Expression of adipogenic transcription factor targets and enrichment of autophagy terms during the course of differentiation.** Gene expression profiles of differentiating adipocytes were curated from previously published data as described in Fig 1. We curated another dataset of publicly available ChIP-seq samples in MDI-induced 3T3-L1 at different time points (Table 2). Read counts from the RNA-seq were used to estimate gene expression. Binding peaks from the ChIP-seq were used to identify the transcription factors binding sites and targets. Differential expression analysis was applied to quantify the change of autophagy gene expressions (fold-change) at different time points compared to pre-adipocytes. The median fold-change of A) PPARG or B) CEBPB targets at the corresponding time points (blue, low & red, high). C) Genes were ranked descendingly rank based on the fold-change. Enrichment of the autophagy "regulation," "selective," and "subtypes" terms was calculated by quantifying the over-representation of their members in the top or bottom ranks of the list (blue, negative & red, positive enrichment).

were enriched at the same time point (48 hr). Terms in the autophagy subtypes that related to the same organelles were also enriched in the up-regulated genes in late-adipogenesis. Together, the biphasic response of autophagy to the MDI-induction was significant in terms of the number of regulated genes and at the gene set level. In particular, selective and organelle-specific forms of autophagy were activated in late-adipogenesis.

## Co-regulators are recruited to ubiquitously bound autophagy gene regions and redistribute over time

We further explored the combined binding of key adipogenic factors and co-factor at genome-scale using a ten-states model of the chromatin at the early and later stages of differentiation. During the early stages of differentiation (4 hours), regions of the chromatin fell into one of two categories demarcated by binding patterns (S1A Fig). The first were either devoid of binding proteins (90%), insulated by CTCF (1%), or repressed by NCOR1 (1%) regardless of the presence of other proteins. These areas were generally stable and mainly transitioned to other

states of the same category. The active areas were ubiquitously bound to multiple proteins, co-factors only, or a specific transcription factor with its known co-factor. CEBPB associated with MED1 (emission probability (EP) = 0.56 and 0.46, respectively) and PPARG associated with RXRG (EP = 0.86 and 0.85, respectively). These regions were likely to make the transition either from similar binding patterns or from areas devoid of factor binding. In sum, regions that are not open for binding remain so. The binding of transcription factors is sometimes associated with insulators or repressors and is mostly accompanied by co-factors.

The same patterns of transcription factor and co-factor combinations in an eight-states model emerged at the later stages of differentiation with notable additions (S1B Fig). Repressed regions were more stable and less likely to transition to other states. Transcription factors CEBPB and PPARG associated with more than a single co-factor. In the case of PPARG, the complex of the transcription factors and co-factors (RXRG and MED1) made the transition from the earlier (PPARG + MED1) or PPARG alone state. However, the CEBPB complex made the transition to areas devoid of factors. Possibly, co-factors allow in, or themselves are being recruited by transcription factors to regions with high binding affinities.

Significant changes in the states that pertain to insulation, repression, or the binding availability of the chromatin occurred in early adipogenesis. On both autophagy and lipogenic genes, the frequency of the chromatin regions ubiquitously bound to regulatory proteins and co-factors in particular increased (> 3 fold) (Fig 3A). This was also accompanied by reduced binding to insulators and repressors (> 2 fold). However, in the longer course of differentiation, the more pronounced changes in state frequency involved the combinations of short and long-acting transcription factors and their association with specific co-factors (Fig 3B). Fewer regions were available for the CEBPB/MED1 complex, and more were available for PPARG either alone or in association with RXRG and MED1.

## Co-factors preceded their factor on the shared targets

To examine the co-occurrence of the adipogenic transcription factors on autophagy, we analyzed the intersections of the features they bind to at different time points. PPARG targeted the largest numbers of autophagy genes (Fig 4A). Those targets localized in the later time points. By contrast, CEBPB had the largest number of targets in pre-adipocytes and early after induction with MDI. The downstream targets of PPARG overlapped with those of RXRG, while CEBPB targets overlapped with MED1, especially in early time points. Moreover, co-factors such as MED1 and RXRG accessed their targets independent of the time point. This is confirmed by calculating the fraction of overlap between the PPARG and CEBPB binding with that of the co-factors (Fig 4B). Unlike CEBPB, the overlap between the targets of PPARG and the co-factors increased over time.

## Co-factors localize to and prime gene enhancers

We then constructed a multi-state chromatin model of histone modifications and examined their co-occurrence with different individual factors and the combinations. Chromatin regions fell into one of two general categories: active or repressed chromatin (S2A Fig). The combination of H3K27ac and H3K4me1 marks the enhancer regions (EP > 0.7 and 0.9), while the combination of H3K27ac and H3K4me3 (EP > 0.8 and 0.9) marks the active promoters. H3K36me3 marks regions with strong transcriptional activity. The enhancers were further classified into active, weak, or genic depending on the distribution of the histone markers. Active regions mainly transitioned within the same category of enhancers and genic enhancers to robust transcription. The inactive chromatin was annotated by either H3K27me3 (Repressed polycomb), H3K9me3 (Repeats), or devoid of any markers (Heterochromatin).

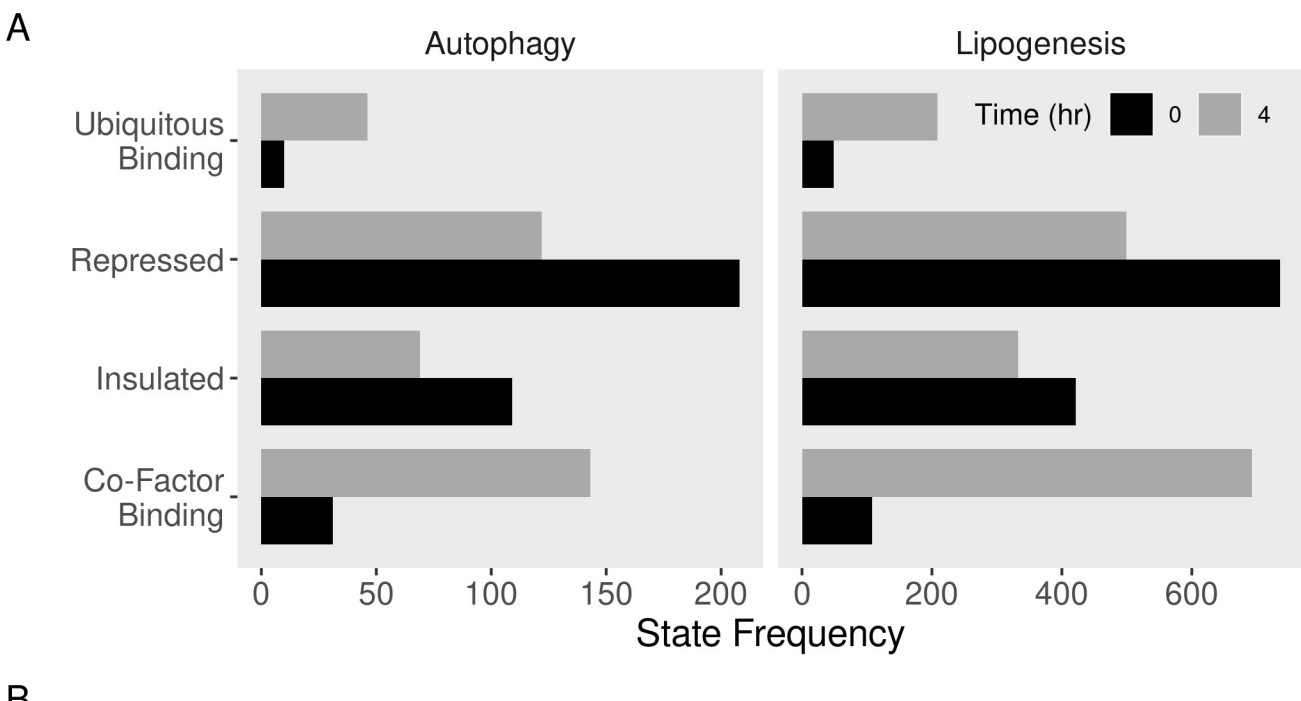

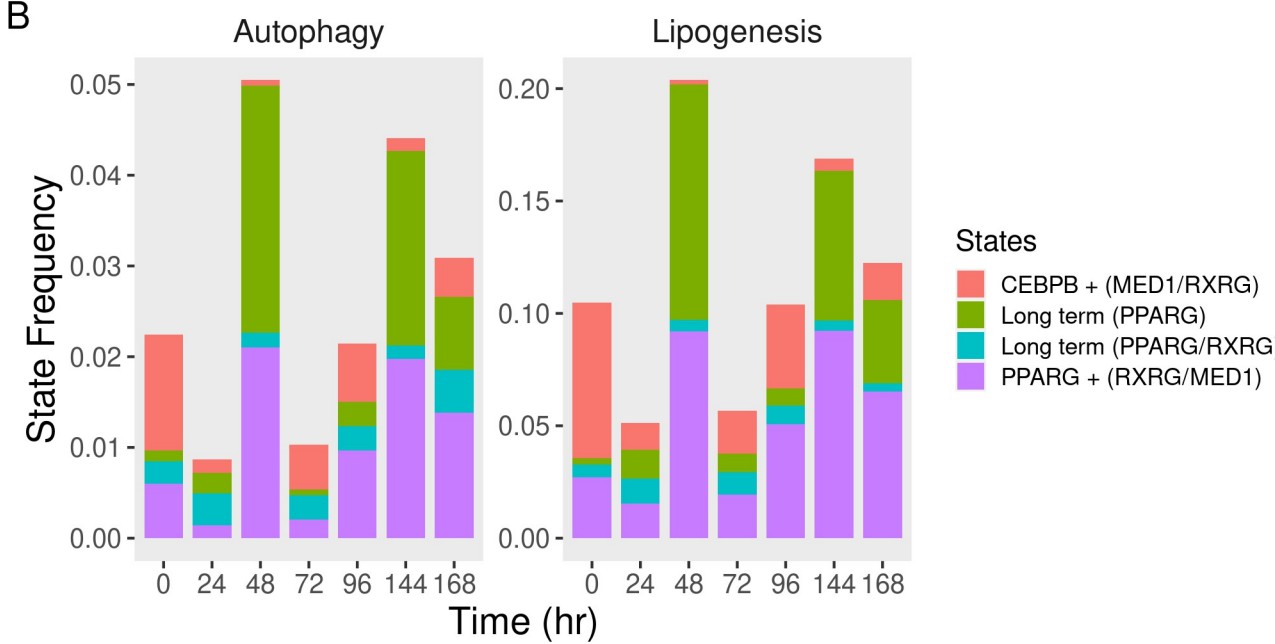

**Fig 3. Frequencies of transcriptional states of autophagy gene regions in differentiating adipocytes.** Multivariate chromatin models were built using binarized signal tracks of transcription factors, co-factors, and DNA-binding proteins in differentiating adipocytes as described in S1 Fig. Autophagy and lipogenesis genomic regions were segmented and labeled by the corresponding states. Frequencies of selected states were calculated at each time point of the A) early-stage (up to 48 hr) and B) full course (up to 10 days) of differentiation.

The only transition from inactive to active regions occurred between weak enhancers and repeats.

Autophagy genes regulatory regions such as promoters and enhancers were enriched in different sets of binding sites (S2B Fig). The active promoter chromatin state was enriched in

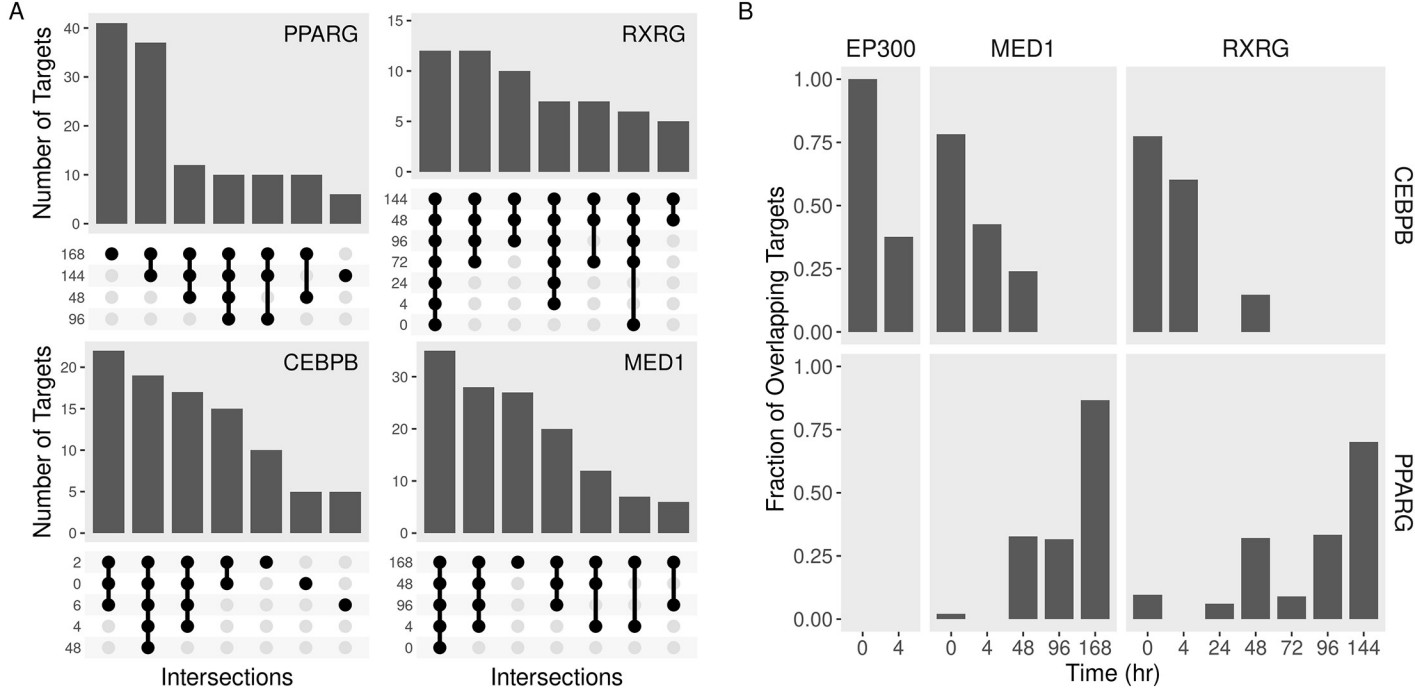

**Fig 4. Autophagy gene target of adipogenic regulators and their overlap at different time points.** Transcription factors (PPARG and CEBPB) and co-factors (RXRG, MED1, and EP300) binding peaks were identified from a curated dataset of publically available ChIP-seq samples of differentiating adipocytes. Binding peaks at each time point were assigned to the nearest autophagy gene. A) The intersection between the targets of every regulator at different time points was calculated. The intersecting sets are shown as connected dots and the sizes of the sets are shown as bars. B) The overlap between the targets of every regulator at different time points was calculated. The fraction of the overlapping targets of the transcription factors and cofactors are shown as bars.

PPARG binding sites independent of the time point (Score = 37–55). These regions were enriched in CEBPB binding sites to a lesser extent (score = 16). Active and genic enhancer states were the most enriched in CEBPB binding sites (score = 18–30). Different enhancer states were enriched in co-factors EP300, MED1, and RXRG binding sites (score = 5–30). The enrichment of enhancers in these binding sites increased over time. These observations suggest that the co-factors may not share the same sites but bind to other regulatory regions of the same target. This might explain the discrepancies between the factor-co-factor overlap based on binding sites vs. gene targets.

Adipogenic transcription factors regulate autophagy through other transcription factors and kinases. PPARG targeted DNA-binding transcription factors, especially early on during the differentiation course (Fig 5A). We first observed that PPARG target several genes labeled as transcription factors and autophagy-related in the gene ontology terms. The effect was more significant (ratio = 0.3 at FDR < 0.2) in the case of autophagy compared to lipogenesis. CEBPB targeted a smaller number of these factors. Both factors targeted genes coding for protein kinases throughout the course of differentiation (Fig 5A). By contrast, the two factors, and CEBPB in particular, increasingly targeted genes coding for protein phosphatases related to lipogenesis. The expression of autophagy transcription factors and kinases coding genes was, on average, induced during the differentiation course (Fig 5A). The knockdown of *Pparg* in pre-adipocytes resulted in failed induction of this set of targets. This effect persisted for several days after the beginning of differentiation. A similar effect was observed for the knockdown of *Cebpb* at 4 hours.

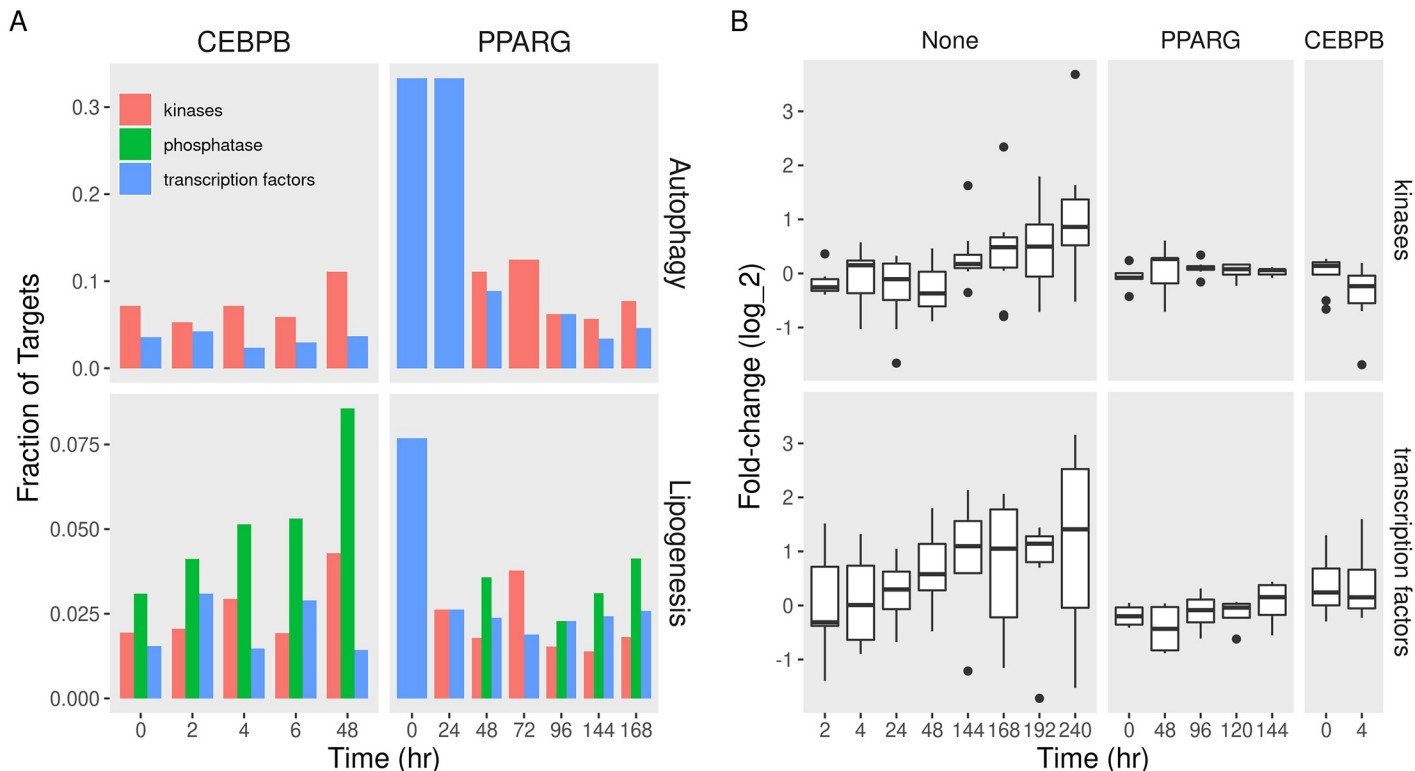

**Fig 5. Fractions and expression of autophagy and lipogenesis gene targets in different functions during adipogenesis.** A) Publicly available PPARG or CEBPB ChIP-seq data in differentiating adipocytes were used to identify autophagy (and lipogenesis) gene targets at every time point differentiation. Gene ontology was used to identify the molecular functions of these targets. The numbers of transcription factor binding targets belonging to a given molecular function are shown as bars. B) Curated datasets of RNA-seq and microarrays of MDI-induced adipocytes with genetic perturbations (PPARG or CEBPB knockdown, Table 4) or without (None, Table 1) were used to estimate gene expression. Differential expression was used to identify the difference in gene abundance between the time points and pre-adipocytes or between knockdown and control conditions. Fold-changes of the genes in each molecular function gene ontology terms in (None) perturbed course, Pparg-or Cebpb-knockdown cells are shown as boxplots (25, 50, and 75% quantiles).

## Discussion

In this study, we used gene expression and chromatin binding data to build models for gene regulation and transcription factors sites/targets in differentiating adipocytes. We were able to identify likely targets among autophagy and lipogenic genes and evaluate the effect of their binding on expression. We also built multi-state models for the transcriptional regulators and chromatin states to explore the interactions between transcription factors, co-factors, and histone modifications. Fig 6 shows a graphical summary of the main findings in the study.

We found that autophagy genes are regulated as part of the transcriptional program of differentiating adipocytes. Therefore, they might be regulated by the same adipogenic transcription factors (Fig 1). We previously made a similar observation [3]. Studies suggested several one-to-one links between those transcription factors. CEBPB induces the expression of *Pparg* either directly or by removing its inhibitors through autophagy [1]. We previously showed that adipogenic transcription factors CEBPB and PPARG regulate autophagy gene products during adipogenesis, either directly or indirectly through other transcription factors. Here, we further explore this regulation by examining the temporal and the spatial arrangement among those two factors, co-factors, and histone modifications.

Autophagy is essential for adipocyte differentiation. The knockdown of crucial autophagy genes such as *Atg5/7* resulted in failed induction of pre-adipocytes or reduced adipose mass

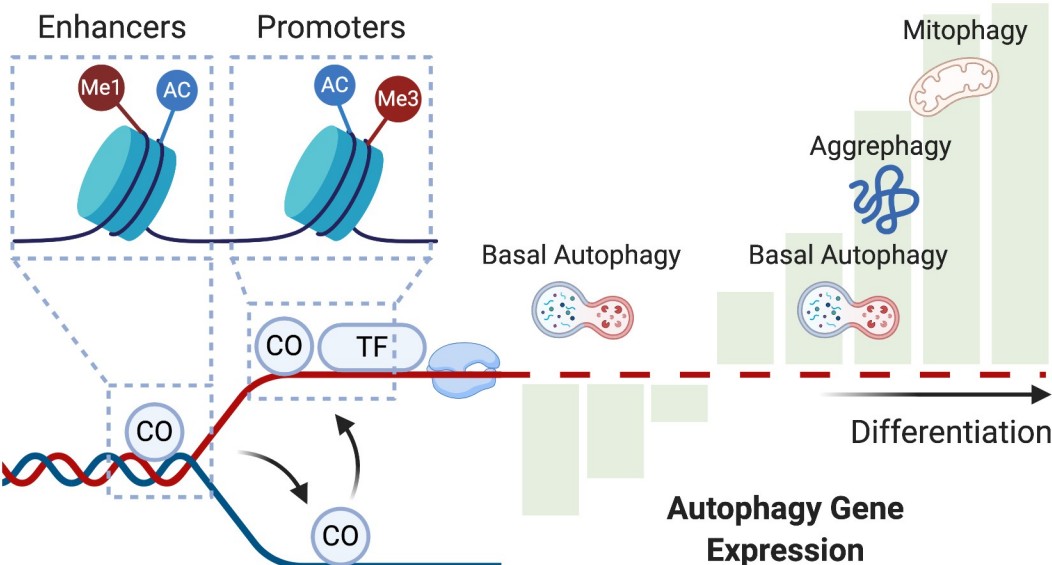

**Fig 6. A model for transcriptional and chromatin modification on autophagy genes.** Co-factors (CO) precede transcription factors (TF) to their shared targets. Co-regulators localize to enhancer regions marked by lysine monomethylation (Me1) and acetylation (Ac). They prime the targets for transcription, where transcription factors bind to the promoter regions marked by lysine tri-methylation (Me3) and acetylation (Ac). These events regulate the expression of autophagy genes in a bi-phasic manner. Early during adipogenesis, several autophagy genes are down-regulated, and possibly only basal autophagy is functional. Later, autophagy genes are up-regulated, and autophagy, organelle-specific autophagy, in particular, is activated.

tissue in mice [22, 23]. We observed the down-regulation of many autophagy gene products in the early days of adipogenesis (Fig 1). This is likely to impede many, but not all autophagy functions. Autophagy plays a role in maintenance and energy production in growing early adipocytes and possibly benefit the white adipocyte phenotype above others [24, 25]. In the later stages of differentiation, cells undergo phenotype changes that require the removal and recycling of intracellular components such as the mitochondria. Indeed, we observed the activation of organelle-specific forms of autophagy after two days of adipocyte induction (Fig 2C). Together, the observed bi-phasic response to MDI-induction suggests two distinct autophagy functions in early and late-adipogenesis.

Although their expression increase in response to MDI, adipogenic factors have binding sites in pre-adipocytes. CEBPB activates as early as 4 hours, and PPARG follows later during the differentiation course [26]. The abundance of the factors during the differentiation might explain this. The former is induced very early during adipogenesis, while PPARG levels do not max out until later. Co-factors exist on their targets irrespective of time points (Fig 4A). They might be able to access the majority of their targets at all times. The combination of factor-cofactor increased overtime for PPARG (Fig 4B). This is either because the complex is binding to more targets over time or because a combination of two proteins is necessary to induce the same targets.

Factors and co-factors localized to different genomic regions, even on the shared targets. As expected, transcription factors CEBPB and PPARG bind to the promoter regions the most. These regions were increasingly modified by histone markers associated with active promoters. PPARG could bind as a single factor suggesting a pioneering function [27]. In other words, it can access DNA, and other factors might provide selectivity. Co-factors were abundant in regions identified as promoters or enhancers (S1 Fig). This suggests that co-factors

such as RXRG and MED1 are required to bind with the main factors but may perform other roles. That could be a form of assisted loading or priming enhancer regions for transcription factors binding [28].

Although RXRG was reported to function as a transcription factor, in many cases, it was also reported to work in partnership with PPARG. We do not hold strong views on the distinction between the factors and co-factors. For the purposes of this study, we treat either by their previously reported functional role. Since the study includes only a handful of regulators, several others must be at work which probably plays a role in the DNA-binding. ChIP-seq data can show that a particular protein is associated with a piece of DNA but doesn't exclude other proteins on the exact site or specific form of direct or indirect binding. Our observation was, for many autophagy gene regions, co-factors appeared to associate the DNA earlier in time than the transcription factor. Those were not necessarily the same binding regions, but regions assigned to the same gene.

A breakdown of the types of binding targets for the two adipogenic factors revealed an interesting pattern. PPARG targeted genes coding for other transcription factors with downstream autophagy targets (Fig 5A). This was also evident in the case of lipogenic genes. CEBPB, on the other hand, was mostly bound to genes involved in other activities such as kinases and proteases. In addition to the larger number of targets, the high number of transcription factors of PPARG suggests a broader effect on regulating autophagy genes. The more downstream transcription factor genes, the larger the effect of the factor. Indeed, knocking down *Pparg* resulted in a broader range of dysregulation (Fig 5B).

Studies suggested that a specific arrangement of transcriptional regulators is required for successful reprogramming of differentiating neurons or neutrophils [29, 30]. Understanding the role of these regulators enables managing the differentiation process and the function of differentiated cells. By manipulating certain factors, it would be possible to fine control the course of cell development. For example, inhibiting mitophagy in pre-adipocytes maintained the beige adipocyte phenotype, rather than the white, and resulted in cells with greater thermogenic capabilities [25]. Reversing the differentiation of adipocytes would also be possible either by targeting the regulators directly or the specific autophagy function they regulate [31]. Finally, mature adipocytes specialize in storing lipids, a function which lipophagy could modify. These observations are only valid insofar as the phenotypes reflect the underlying gene expression. We used the gene set enrichment and over-representation analysis to quantify the changes at the gene set level, which is more likely to correspond to biological functions than claims based on changes in individual genes.

Our analysis was limited to the time points for which data was available. For example, we observed that CEBPB effectively targeted and affected the expression of autophagy genes as early as 4 hours. No data before 4 hours were available. Therefore, we do not know whether this effect can be observed earlier. Our analysis doesn't rule out the involvement of other transcription factors in regulating autophagy genes. We observed significant binding and gene expression changes during adipogenesis that seem to be correlated with those two factors. In addition, the two factors and their co-factors explain a significant portion of the variance over time.

The curated datasets comprise data from previously published studies. This noise may result in masking key findings or exaggerating the effect size of others. We carefully curated and processed the data to reduce batch effects resulting from the variations among the studies. In the case of ChIP-seq data, we only used replicated peaks when more than one sample was available. The integration of more than two data types (RNA-Seq and ChIP-seq) necessitate careful matching of the biological, genetic metadata to make sure the samples correspond to parallel conditions and the information on the expression and the binding matches that corresponding entities. *Pparg*-knockdown microarray data had missing information on multiple

genes that did not have corresponding probes. Finally, in the factor perturbation data during the differentiation course, it was difficult to disentangle the time from the perturbation effect.

The findings presented in this manuscript were based on publicly available data of differentiating adipocytes with or without perturbation. We used quantitative methods such as differential expression, gene set enrichment and chromatin segmentation to support the observations presented in the manuscript. Some of the findings were based a correlation between two or more observations. Those were not always possible to verify using the available data and so remain to be experimentally tested.

## Supporting information

**S1 Fig. Transcriptional regulators multi-state model in early and full differentiation courses.** We curated a dataset of publicly available transcription factors, co-factors, and DNA-binding proteins ChIP-seq samples in MDI-induced 3T3-L1 at different time points (Table 2). Binarized binding signals were used to indicate the presence or absence of regulators at 100 bp windows of the chromatin. A multi-variate model of combinations of regulators was built to summarize A) ten states in the early (up to 48 hr) stage or B) eight states in the full course (up to 10 days) of differentiation. Emission, the probability of each marker being at a given state. Transition, the transitional probability of a given state from/to another. white, low, and black, high probability.
(PNG)

**S2 Fig. Histone modifications multi-state model and overlap with adipogenic regulators.** We curated a dataset of publicly available histone modification ChIP-seq samples in MDI-induced 3T3-L1 at different time points (Table 3). Binarized binding signals were used to indicate the presence or absence of histone markers at 200 bp windows of the chromatin. A multi-variate model of combinations of markers was built to summarize nine states during the course of differentiation. A) Emission, the probability of each marker being at a given state. Transition, the transitional probability of a given state from/to another. white, low, and black, high probability. B) Binding peaks of adipogenic regulators were used to identify gene targets at different time points of adipocyte differentiation from publicly available ChIP-seq sample. The fractions of overlap between the regulators' binding sites and the chromatin states are shown as color values (white, low, and black, high).
(PNG)

## Acknowledgments

We thank all the lab members for their constructive feedback on the study.

## Author Contributions

**Conceptualization:** Mahmoud Ahmed, Deok Ryong Kim.

**Data curation:** Mahmoud Ahmed.

**Formal analysis:** Mahmoud Ahmed.

**Funding acquisition:** Deok Ryong Kim.

**Supervision:** Deok Ryong Kim.

**Validation:** Mahmoud Ahmed.

**Writing – original draft:** Mahmoud Ahmed.

**Writing – review & editing:** Trang Huyen Lai, Trang Minh Pham, Sahib Zada, Omar Elashkar, Jin Seok Hwang, Deok Ryong Kim.

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
