## [Decision Letter · Decision Letter 0]

8 Jul 2021

PONE-D-21-12883

Hierarchical Regulation of Autophagy During Adipocyte Differentiation

PLOS ONE

Dear Dr. Deok Ryong Kim,

Thank you for submitting your manuscript to PLOS ONE. After careful consideration, we feel that it has merit but does not fully meet PLOS ONE’s publication criteria as it currently stands. Therefore, we invite you to submit a revised version of the manuscript that addresses the points raised during the review process.

Based on the two reviewers comments, it seems that this manuscript has utilized datasets from your previous publications. Therefore it is important to address how this study is different from the previously published ones. 

We look forward to receiving your revised manuscript.

Kind regards,

Naresh Doni Jayavelu, Ph.D

Academic Editor

PLOS ONE

Journal Requirements:

Reviewers' comments:

Reviewer's Responses to Questions

**Comments to the Author**

1. Is the manuscript technically sound, and do the data support the conclusions?

Reviewer #1: Partly

Reviewer #2: No

2. Has the statistical analysis been performed appropriately and rigorously? 

Reviewer #1: Yes

Reviewer #2: No

3. Have the authors made all data underlying the findings in their manuscript fully available?

Reviewer #1: Yes

Reviewer #2: No

4. Is the manuscript presented in an intelligible fashion and written in standard English?

Reviewer #1: Yes

Reviewer #2: No

5. Review Comments to the Author

Reviewer #1: In their manuscript, “Hierarchical Regulation of Autophagy During Adipocyte Differentiation,” Ahmed and colleagues has attempted to identify the order of autophagic event during adipocyte regulation. To do this, they made use of the datasets that are previously available. The manuscript reports a series of temporal and spatial organization between adipogenic transcription factors (TFs) and co-factors in the regulation of autophagy-related gene expression by integrating multiple high-throughput sequencing data including RNA-seq, histone ChIP-seq and TF ChIP-seq data.

On the one, the efforts and steps taken to identify the series of events is highly appreciated - is extremely important and interesting. On the other hand, the paper is rather disappointing, because the authors did not attempt to validate the prediction. However, I think if the premise of the paper could be strengthened by either

1) assessing functional outcome of the prediction, for example, studies to correlate autophagy with gene expression.

(or)

2) I understand these studies could be time consuming, in that case I suggest providing the separate limitation section and alter the language throughout by not making any strong claims.

As one of the reviewer pointed out early, Fig2A, B and 6A are hard to understand. I suggest authors to come up with different way of representing the data.

Minor corrections:

All the figure legends repeat the same first sentence again and again. Legends could benefit from detailed info. Also, the legends give a feel that the actual screens were performed in this paper. Legends need serious attention.

Too many figures in the main text. Consider moving some of them as supplementary.

Finally, the manuscript could benefit from rigorous language editing.

Reviewer #2: As have been reported and analyzed in prior work by the same group (https://doi.org/10.3390/cells8111321, https://doi.org/10.1080/21623945.2019.1697563, https://doi.org/10.1080/21623945.2020.1829852), all the gene expression and the DNA-binding events have already been displayed using the same datasets. The present study does not provide anything new and exciting. Also I agree with the following comments that the autophagy gene regulation might not be as specific as the authors stated as thousands of genes will be indcued during adipogenesis by adipogenic transcription factors. I doubt why the authors only choose Cebpb and Pparg, not Cebpa and Fabp4, which are also important. Most importantly, the different autophagy stages are not clear as the authors have no experimental data for support. In conclusion, the idea maybe interesting, but the data is too old and the model is not supportive.

6. PLOS authors have the option to publish the peer review history of their article (what does this mean?). If published, this will include your full peer review and any attached files.

Reviewer #1: No

Reviewer #2: No

---

## [Author Response · Author response to Decision Letter 0]

10 Sep 2021

Reviewer #1:

In their manuscript, “Hierarchical Regulation of Autophagy During Adipocyte Differentiation,” Ahmed and colleagues has attempted to identify the order of autophagic event during adipocyte regulation. To do this, they made use of the datasets that are previously available. The manuscript reports a series of temporal and spatial organization between adipogenic transcription factors (TFs) and co-factors in the regulation of autophagy-related gene expression by integrating multiple high-throughput sequencing data including RNA-seq, histone ChIP-seq and TF ChIP-seq data.

On the one, the efforts and steps taken to identify the series of events is highly appreciated - is extremely important and interesting. On the other hand, the paper is rather disappointing, because the authors did not attempt to validate the prediction. However, I think if the premise of the paper could be strengthened by either

1. assessing functional outcome of the prediction, for example, studies to correlate autophagy with gene expression. (or) I understand these studies could be time consuming, in that case I suggest providing the separate limitation section and alter the language throughout by not making any strong claims. As one of the reviewer pointed out early, Fig2A, B and 6A are hard to understand. I suggest authors to come up with different way of representing the data.

We expanded the limitations part of the discussion to address the reviewer's concerns. In addition, we edited the figures and the legends to better describe the data presented and how it can be interpreted.

Minor corrections:

• All the figure legends repeat the same first sentence again and again.

We edited the figure legends to better reflect the data shown in them.

• Legends could benefit from detailed info. Also, the legends give a feel that the actual screens were performed in this paper. Legends need serious attention.

We edited the figure legends to include detailed descriptions of the data and analysis presented in each.

• Too many figures in the main text. Consider moving some of them as supplementary.

We reduced the number of figures by using more compact representations and moving two figures to supplementary materials

• Finally, the manuscript could benefit from rigorous language editing.

We revised the language in the manuscript to correct grammatical mistakes and typos and to improve clarity. We also plan to use professional editing service to revise the manuscript before the publication.

Reviewer #2:

As have been reported and analyzed in prior work by the same group (https://doi.org/10.3390/cells8111321, https://doi.org/10.1080/21623945.2019.1697563, https://doi.org/10.1080/21623945.2020.1829852), all the gene expression and the DNA-binding events have already been displayed using the same datasets. The present study does not provide anything new and exciting.

In a previous study, we showed that two important adipogenic transcription factors (PPARG and CEBPB) regulate the expression of autophagy-related genes either directly or indirectly through other specific transcription factors. In this study, we investigated the spatial and temporal arrangement of those two factors and their cofactors. We believe that these findings elaborate on the mechanisms of the transcription regulation of autophagy by these transcription factors and co-factors. In addition, we showed that these regulatory links are potentially involved in the induction of specific forms of autophagy, have a strong temporal dimension, and work through specific effectors (other factors or kinases) compared to the regulation of lipogenesis. In response to the reviewers' comments, we revised the Introduction & Discussion section to stress the value added by the study to previous work.

Also I agree with the following comments that the autophagy gene regulation might not be as specific as the authors stated as thousands of genes will be indcued during adipogenesis by adipogenic transcription factors.

The reason we believe autophagy genes are regulated by adipogenic transcription factors is the following

1. Previous studies have shown that knocking out ATG5/7 in preadipocytes resulted in failure of differentiation.

2. The expression of autophagy genes changed over the course of differentiation

3. We found binding sites of PPARG and CEBPB on autophagy genes

4. The knockdown of PPARG or CEBPB in preadipocytes resulted in an autophagy gene expression signature that is different during the course of differentiation compared to control cells.

Our initial goal was to study the regulation of autophagy during adipogenesis mainly through gene expression data. In previous work, we found that autophagy genes are regulated as part of the transcriptional program and adipogenic transcription factors and co-factors were the obvious candidates as drives of the gene expression. As a result, we used ChIP-seq data to identify binding patterns and correlate them with changes in gene expression. The role of these factors is well described in the case of the processes that are directly related to adipogenesis and lipogenesis. Therefore, we believe it is useful to study these patterns of regulation in the case of autophagy. We also found key differences between the way autophagy and lipogenesis genes are regulated by these factors. Namely, the autophagic response to the adipogenic induction is biphasic and the adipogenic factors target other factors and kinases more specifically. We clarified the motivation for the study and the focus on autophagy genes in the Introduction section.

I doubt why the authors only choose Cebpb and Pparg, not Cebpa and Fabp4, which are also important.

Our analysis doesn't rule out the involvement of other transcription factors in regulating autophagy genes. We observed significant binding and gene expression changes during adipogenesis that seem to be correlated with those two factors. In addition, the two factors and their co-factors explain a significant portion of the variance over time. We discussed this issue in the limitations of the study in the Discussion section

Most importantly, the different autophagy stages are not clear as the authors have no experimental data for support. In conclusion, the idea maybe interesting, but the data is too old and the model is not supportive.

Our analysis is based on two data types (RNA and ChIP-seq). We reported observations on the changes in gene expression, DNA-binding, and the correlation of the two. We don't make any explicit claims on the function, since our analysis doesn't include any functional data. We recognize that this kind of observation is only valid insofar as phenotypes reflect the underlying gene expression. We used the gene set enrichment and over-representation analysis to statistically quantify these observations at the gene set level, with is more likely to correspond to biological functions than claims based on changes in the expression of individual genes. We explained this view with the other limitations of the study in the Discussion section.

---

## [Editor Report · Decision Letter 1]

6 Dec 2021

Hierarchical Regulation of Autophagy During Adipocyte Differentiation

PONE-D-21-12883R1

Dear Dr. Deok Ryong Kim,

We’re pleased to inform you that your manuscript has been judged scientifically suitable for publication and will be formally accepted for publication once it meets all outstanding technical requirements.

Kind regards,

Naresh Doni Jayavelu, Ph.D

Academic Editor

PLOS ONE

Additional Editor Comments (optional):

Authors sufficiently addressed both reviewers comments and added limitation of the current study. I think it is ready for publication now.
---

## [Editor Report · Acceptance letter]

31 Dec 2021

PONE-D-21-12883R1 

Hierarchical Regulation of Autophagy During Adipocyte Differentiation 

Dear Dr. Kim:

I'm pleased to inform you that your manuscript has been deemed suitable for publication in PLOS ONE. Congratulations! Your manuscript is now with our production department. 

Kind regards, 

on behalf of

Dr. Naresh Doni Jayavelu 

Academic Editor

PLOS ONE